# Dynamic impact of E-WOM and green discourse on green hotel supply chain performance with time lag effect

Lujie Hao[1], Bingkun Lin[2]*, Yaping Zhu[2]

1 International Business School, Fuzhou University of International Studies and Trade, No. 28, Yuhuan Road, Shouzhan New District, Changle District, Fuzhou City, Fujian Province, PR China, 2 Business School, Minnan Normal University, No. 36, Xianqianzhi Street, Xiangcheng District, Zhangzhou City, Fujian province, PR China

* linbk99@163.com

## Abstract

This paper investigates a signal online channel green hotel supply chain comprising a green small and medium-sized hospitality enterprise (GSMHE) responsible for offline electronic Word-of-Mouth (e-WOM) promotion and green service efforts, and an online travel agency (OTA) engaged in online e-WOM promotion activities. Both offline and online e-WOM efforts significantly contribute to the overall e-WOM level of the green hotel supply chain. Due to the inherent characteristic of reviews posting, the impacts of these efforts on e-WOM exist time lags. Therefore, this study aims to explore the influence of time lags and examine the feasibility of a cooperative program in this context. This paper develops a differential game model that incorporates time lags. By analyzing and comparing the equilibrium strategies and profits under three scenarios, we demonstrate that a cost-sharing model can be successfully implemented when specific relationships regarding the time lags of the GSMHE are satisfied.

## 1. Introduction

In recent years, customers have shown a growing awareness of environmentally conscious practices, which has led to a stronger preference for eco-friendly products and services [1]. In the tourism industry, there has been a steady rise in travelers' demand for environmentally sustainable accommodations [2]. Official reports indicate that over 92% of consumers express positive attitudes toward businesses that prioritize environmental protection [3]. In response to the increasing environmental consciousness among potential customers, hoteliers have dedicated significant efforts to developing and promoting eco-friendly practices as a means to gain a competitive advantage, foster guest loyalty, and ensure regulatory compliance [4–6]. For

**Data availability statement:** All relevant data are within the paper and its Supporting information files.

**Funding:** LH acknowledges the support from the Social Science Foundation of Fujian (grant numbers FJ2023B059 and FJ2025C041) and the Fuzhou Key Research Institute of Social Science Foundation (grant number 2024FZB32).BL acknowledges the support from the Social Science Foundation of Fujian (grant number FJ2025B165). The funders had no role in study design, data collection and analysis, decision to publish, or preparation of the manuscript. There was no additional external funding received for this study.

**Competing interests:** The authors have no relevant financial or non-financial interests to disclose.

instance, The Grand Hotel in Birmingham has successfully implemented eco-friendly practices, attracting more sustainability-conscious travelers [7].

Despite these efforts, hotel managers face considerable challenges in effectively communicating their green initiatives to potential customers [8]. With the internet now serving as the primary tool for travelers seeking accommodations, studies emphasize its effectiveness as a direct marketing channel for promoting green initiatives [9,10]. However, the credibility of these channels remains critical for environmentally conscious travelers [11]. As a result, the green hospitality sector increasingly turns to electronic word-of-mouth (e-WOM), which research has shown to be more credible and trustworthy than traditional advertising, significantly influencing consumer decision-making [12,13]. Consequently, potential guests now rely heavily on online reviews (e-WOM) to form their perceptions and make informed decisions about green accommodations [12,14].

Nevertheless, the effectiveness of direct marketing channels varies across the hospitality industry, including both large hotel chains and small and medium-sized hospitality enterprises (SMHEs) [15]. While larger hotels have successfully adopted these strategies, SMHEs often struggle with smaller marketing budgets, limiting their investment in e-WOM marketing [16]. Thus, many SMHEs rely on Online Travel Agencies (OTAs) for marketing and distribution [17,18]. OTAs also offer platforms for e-WOM, enabling SMHEs to promote their green image and sustainable practices [19].

Despite considerable scholarly attention to environmental sustainability in SMHEs [20–22], limited research has explored how SMHEs utilize OTAs to promote green initiatives. Moreover, the role of e-WOM within the green hotel supply chain remains underexplored. It is necessary to establish a mechanism that fosters collaboration between SMHEs and OTAs to promote e-WOM in the context of green initiatives. To address these issues, our study introduces a novel approach by focusing on a single supply chain structure led by a green SMHE, referred to as GSMHE, in collaboration with an OTA. We employ a Stackelberg differential game framework integrating e-WOM and green discourse to model decision-making dynamics in the Green Hotel supply chain. The OTA assumes the leader role, determining its own online e-WOM effort and cost-sharing rate for the GSMHE. The GSMHE acts as the follower, making decisions regarding its own offline e-WOM effort and green service provision. Furthermore, we examine the influence of e-WOM on decision-making, with particular attention to time delays in both online and offline e-WOM promotion efforts, a factor that has not been fully addressed in previous studies. Through backward induction, we derive equilibrium solutions and compare performance across scenarios: online e-WOM promotion, online and offline e-WOM promotion cooperation, and cost sharing. The primary objectives of this paper are as follows: (1) To determine the optimal levels of green service effort for the GSMHE and e-WOM efforts for both the GSMHE and OTA. (2 To assess how time lags in e-WOM affect these efforts, the trajectory of e-WOM, and the overall profitability of both the GSMHE and OTA (3) To identify the scenario that yields best performance in terms of profitability for both the GSMHE and OTA by considering the time lags.

The subsequent sections of this paper are organized as follows. Section 2 provides a comprehensive review of the relevant literature. Section 3 presents the formulation of a differential game model. Section 4 focuses on comparing the results across three scenarios and sensitivity analysis. Furthermore, Section 5 presents a numerical example. Finally, Section 6 concludes the paper by summarizing the key findings, addressing implications.

## 2. Literature review

### 2.1 Hotels and OTA in hotel supply chain

The growth of OTAs in the digital era has revolutionized tourism bookings, including hotel rooms and flights, due to economic development and technological advancements [23]. Given the intangible nature of tourism products, customers often engage in offline experiences after making online reservations. This has led to the widespread adoption of the Online-to-Offline (O2O) model, where OTAs and hotels collaborate to enhance the customer experience [24]. The introduction of OTAs has shifted the traditional "Hotel-Consumer" relationship into a more dynamic "Hotel-OTA-Consumer" interaction [25], benefiting SMHEs by providing access to a broader customer base, including potential guests who might not have been previously aware of the hotel's offerings [26]. This expanded reach has contributed to increased room sales and higher occupancy rates for SMHEs [27].

Extensive research has explored the complex dynamics between OTAs and tourism operators, focusing on both the competitive and cooperative aspects of their interactions [28–30]. While existing literature acknowledges the existence of channel conflict between hotels and OTAs [26], particularly in their efforts to retain returning customers [28], a growing body of literature advocates for cooperative strategies that maximize profitability for both parties [29–33]. This cooperative approach is rooted in the recognition that OTAs play a crucial role in promoting the hotel brand and facilitating the attraction of potential customers to the hotel [28,34,35].

Moreover, scholars have extensively studied the cooperative interactions between hotels and OTAs within the broader tourism supply chain [36–38]. While hotels prefer to cooperate with OTA with sufficient online customers [35], empirical evidence reveals that factors such as commission rates, hotel size, and OTA channel acceptance influence the cooperation between hotels and OTAs [26]. For instance, there is an inverse relationship between hotel occupancy rates, the number of hotels listed on OTA websites, and the willingness of hotels to engage in cooperative efforts with OTAs [35]. Effective room demand forecasting and availability management are crucial for ensuring profitability in hotel-OTA partnerships [32]. Furthermore, strategic sales initiatives and optimal room reservation practices not only stimulate market demand but also improve occupancy rates, creating mutual benefits for both parties [39]. In this context, revenue-sharing agreements have emerged as widely accepted mechanisms for aligning the interests of hotels and OTAs and achieving effective channel coordination [39,40]. Additionally, advertising fees have been proposed as another cooperative strategy to further strengthen hotel-OTA collaboration [41].

The selection between the merchant and agency models is another area of investigation. OTAs tend to prefer the agency model when hotel's capacity is relatively smaller [40]. Furthermore, when OTAs offer lower commission rates or display increased altruistic preferences, both parties lean towards the agency model [26,42,43]. In contrast, the merchant model is more likely to be adopted when the hotel's capacity is larger [40]. Hotels with limited bargaining power are also more inclined to select the merchant model [44]. Moreover, research has explored optimal pricing strategies, taking into account factors such as service level, service cost coefficient, unit sale commission, and unit service compensation, all of which influence pricing decisions for both OTAs and hotels [29]. The pricing strategy adopted by one party can affect the purchasing decisions of the other, and hotels can shape OTA choices by adjusting wholesale prices and managing overbooking levels [45,46].

The growing focus on "green development" within the tourism industry has led to strengthened collaboration between green hotels and OTAs [47]. Scholars have begun to explore the dynamics of green and sustainable supply chains in this context, highlighting the role of factors such as service quality [38], altruism preference [43,48], procedural fairness

[49], and the utilization of big data [48]. These elements significantly contribute to the sustainability of both hotel and OTA operations. Furthermore, research suggests that revenue-sharing agreements are the optimal contract design for fostering cooperation in green tourism supply chains [47,50].

## 2.2 Green hotels and e-WOM

As green hotels and OTAs collaborate to meet the growing demand for sustainable tourism, consumers' increasing environmental consciousness has further propelled this trend. This rising awareness has led to a surge in demand for sustainable practices within the hospitality industry [51–53], positioning the sector as a vanguard in implementing green initiatives [22,54,55]. The perception of green practices in hotels significantly influences customers' preference for environmentally-friendly accommodations [51,56,57]. To effectively communicate their sustainability efforts, green hotels employ strategies such as brochures, exhibitions, and social media engagement [58,59].

To further enhance credibility and trustworthiness, the green hospitality sector has increasingly embraced e-WOM as a crucial tool in digital marketing strategies [12]. The rise of online reviews (ORs) on digital platforms has made e-WOM increasingly prevalent in the hospitality industry [60]. These reviews provide valuable feedback from customers about their hotel experiences, serving as a critical resource for potential travelers [61]. Scholars widely recognize ORs as a prominent form of e-WOM, significantly influencing how prospective customers perceive service quality and make accommodation choices [62,63]. Research indicates that ORs are particularly effective in evaluating the environmental practices of service providers [63,64]. Consequently, potential guests rely on ORs to assess the "green perception" of hotels based on previous guests' experiences [65]. Positive ORs play a crucial role in influencing customers' intentions to choose green hotels [66,67], particularly those with a strong environmental discourse [60,68].

Based on the literatures, we found that while sustainability remains a key priority in the tourism industry [69,70], the collaboration between green hotels and OTAs presents an opportunity for a win-win outcome [49,50]. However, despite the reliance of SMHEs on OTAs, research on the specific dynamics between GSMHEs and OTAs remains limited. While OTAs provide platforms for consumers to share experiences through online reviews and images [71, 72], much of the literature focuses on the impact of e-WOM on consumer behavior in general, with a notable emphasis on the influence of e-WOM on travelers' decision-making processes [61,66,73]. However, few studies have explored the joint efforts of green hotels and OTAs in promoting e-WOM or examined strategies for enhancing e-WOM within the green hotel supply chain. This gap in literature signals an opportunity to investigate how GSMHEs and OTAs can collaborate more effectively to promote their green initiatives through e-WOM, a crucial aspect in the context of sustainable tourism.

Furthermore, while it is well-established that the promotion of e-WOM is not instantaneous and may involve a delay effect [74], few studies have fully explored the time-lag dynamics within the e-WOM process. Efforts by green hotels and OTAs to enhance their online presence and customer reviews may not produce immediate changes in consumer behavior or brand perception. Instead, the impact of these efforts tends to accumulate over time, as customers gradually engage with the content, share their experiences, and influence the decisions of potential guests [75]. This gradual process emphasizes the time-lagged nature of e-WOM, where the effects of promotional efforts materialize incrementally. While the time lag effect has been explored in other supply chain contexts, such as Corporate Social Responsibility (CSR) investments [76], low-carbon investment [77,78], green technology spillovers [79], product quality and service quality [80], the effect of advertising [81,82], limited research has specifically examined the time-lag effect in the promotion of e-WOM within the green hotel supply chain.

To address these gaps, this paper introduces a novel method that dynamically addresses both online and offline e-WOM by formulating a signaling channel within the green hotel supply chain, composed of GSMHEs and OTAs. In contrast to previous studies, this approach emphasizes that GSMHEs can actively foster offline e-WOM, while OTAs focus on augmenting e-WOM on their platforms. Additionally, acknowledging the time delay involved in the e-WOM promotion within the green hotel supply chain, this research incorporates time-lag factors into the dynamics of e-WOM. This novel

approach not only contributes to advancing theoretical understanding but also provides practical insights into how green hotels and OTAs can optimize their collaborative e-WOM strategies for sustained long-term benefits.

## 3. Model formulation

As GSMHE faces challenges in independently running direct online channels due to low popularity and a lack of e-commerce experience [29], we propose a single green hotel supply chain structure with one GSMHE as the producer and one OTA as the retailer in this section. GSMHE provides hotel services that are sold exclusively through the OTA. We use subscripts "h" and "o" to denote the GSMHE and OTA, respectively. In this green hotel supply chain, the OTA determines online e-WOM effort $E_o(t)$ and cost sharing rate $\Phi$. The GSMHE decides green service effort $S_h(t)$ and offline e-WOM effort $E_h(t)$. This creates a Stackelberg differential game between the leading OTA and the following GSMHE. The operation model is depicted in Fig 1 and the notations used in this paper are presented in Table 1.

Let G(t) denote the discourse level of the GSMHE's green service, where $G_0 \geq 0$ represents the initial discourse level of the green service. As tourists increasingly prioritize environmental concerns and show a preference for green tourism products, hotels actively invest in green services to attract environmentally aware visitors [83]. Scholars have shown that green discourse in online reviews serves as a key indicator of tourists' experiences and evaluations of these green services [84]. Thus, the discourse level of the green service is closely linked to GSMHE's green service efforts. Additionally, green discourse exhibits a decaying pattern over time [84]. Drawing upon the Nerlove-Arrow model [85], we can formulate the dynamics of the green discourse as follows:

$$\dot{G}(t) = \tau_1 S_h(t) - \gamma G(t), \, G(0) = G_0 \tag{1}$$

Where $\tau_1$ denotes an effect parameter of GSMHE's green service effort on green discourse, $S_h(t)$ denotes the efforts GSMHE green service effort at time t, $\gamma > 0$ reflects the decay rate of the green discourse over time.

Let W(t) represent e-WOM, where $W_0 \geq 0$ denotes the initial e-WOM level. The presence of environmental discourse within Online Reviews (ORs) is positively associated with e-WOM [62]. Both GSMHEs and OTAs should actively endeavor to enhance the level of e-WOM by encouraging guests to write ORs and improve the environmental discourse content within them. Conversely, e-WOM exhibits a decay over time, influenced by the occurrence of negative ORs and the decay of green discourse [84]. Drawing on the Nerlove-Arrow lag model [78], the dynamics of e-WOM with a time lag can be formulated as follows:

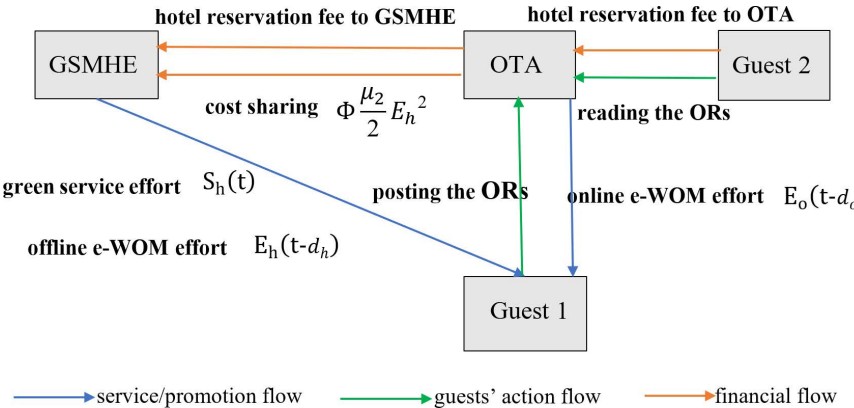

**Fig 1. The research frame.**

$$\dot{W}(t) = \eta G(t) + \tau_2 E_h(t - d_h) + \tau_3 E_o(t - d_o) - \delta W(t), W(0) = W_0 \qquad (2)$$

where $\eta > 0$ is an effect parameter of green discourse on e-WOM. $\tau_2 > 0$ is an effect parameter of the e-WOM efforts of GSMHE on e-WOM, $E_h(t)$ denotes the efforts for e-WOM of GSMHE at time t, $d_h$ represents the lag time of offline e-WOM effort for GSMHE, indicating that GSMHE make efforts produces an e-WOM effect only after a fixed $t - d_h = 0$, which is mainly caused by the guests post reviews on OTA website within the period of time $d_h$; $\tau_3 > 0$ is an effect parameter of the e-WOM efforts of OTA on e-WOM, $E_o(t)$ denotes the efforts for e-WOM of OTA at time t, $d_o$ represents the lag time of e-WOM effort for OTA, indicating that OTA make efforts produces an e-WOM effect only after a fixed $t - d_o = 0$, which is mainly caused by the guests post reviews on OTA website within the period of time $d_o$; $\gamma > 0$ reflects the decay rate of the e-WOM over time. GSMHE engages in offline e-WOM promotion during guests' hotel stays, while OTA focuses on online e-WOM promotion following guests' departure, leading to a time lag where $d_h \geq d_o$.

E-WOM has emerged as an influential marketing tool, as customers rely on the opinions of others to make purchase decisions [86]. Online reviews play a pivotal role in raising hotel awareness, even among unfamiliar customers [87]. Given the uncertainty of assessing service quality before purchase, prospective customers heavily rely on e-WOM from online reviews [61]. Consequently, e-WOM significantly impacts hotel profitability [88]. In addition, the presence of green discourse in online reviews further enhances the influence of e-WOM [60], positively affecting consumers' purchasing intentions in the tourism industry.

Therefore, we assume that market demand for the green hotel supply chain is influenced by green discourse and e-WOM levels. To maintain a clear focus on the impact of these non-price factors, we exclude price considerations from the model, in accordance with the demand function formulation used in similar research [89]. Hence, the demand function in this study is expressed as:

$$D(t) = \alpha G(t) + \beta W(t) \qquad (3)$$

**Table 1. The notation and definitions.**

| Notations | Definition |
|---|---|
| $S_h, E_h, E_o$ | Green service, GSMHE's offline e-WOM promotion efforts, OTA's online e-WOM promotion efforts |
| $\tau_1, \tau_2, \tau_3,$ | Sensitivity of green service to green discourse level; sensitivity of GSMHE's offline effort to e-WOM level; sensitivity of OTA's online effort to e-WOM level |
| $\eta$ | Sensitivity of green discourse to e-WOM level |
| $\alpha, \beta$ | Sensitivity of green discourse and e-WOM to market demand |
| $\gamma, \delta$ | Decay rates of green discourse and e-WOM |
| $\mu_1, \mu_2, \mu_3$ | Sensitivity of green service, GSMHE's offline e-WOM promotion efforts, OTA's online e-WOM promotion efforts to the Cost functions |
| $C(S_h), C(E_h), C(E_o)$ | Cost functions of the GSMHE and the OTA |
| $\rho$ | Discount rates |
| $d_h, d_o$ | Time lags of GSMHE and OTA |
| $\pi_h, \pi_o$ | Marginal revenue of GSMHE and OTA |
| $J_h, J_o$ | Profit functions of GSMHE and OTA |
| $G(t)$ | Green discourse at time t |
| $W(t)$ | E-WOM level at time t |

where $\alpha > 0$, $\beta > 0$, denote the effects of green discourse, e-WOM, on current demand, respectively. The total costs include the green service effort cost of GSMHE, as well as the e-WOM effort costs incurred by both GSMHE and OTA. In line with previous literature, we assume the above costs have convex features. The functions are written as follows:

$$C(S_h) = \frac{\mu_1}{2} S_h^2 \tag{4}$$

$$C(E_h) = \frac{\mu_2}{2} E_h^2 \tag{5}$$

$$C(E_o) = \frac{\mu_3}{2} E_o^2 \tag{6}$$

where $\mu_1 > 0$, $\mu_2 > 0$, $\mu_3 > 0$, denote parameters related to cost of green service effort by GSMHE, e-WOM effort by GSMHE, e-WOM effort by OTA.

The GSMHE's marginal revenue from selling rooms through OTA is denoted as $\pi_h$, while the OTA's marginal revenue from selling rooms is represented as $\pi_o$, where both variables are subject to their respective marginal operating costs. Assuming an infinite time horizon and a discount rate $\rho > 0$, the objective function of the GSMHE is expressed as:

$$J_h = \int_0^\infty e^{-\rho t} \left( \pi_h \cdot D(t) - C(S_h) - C(E_h) \right) dt \tag{7}$$

and the objective function of the OTA is written as

$$J_o = \int_0^\infty e^{-\rho t} \left( \pi_o \cdot D(t) - C(E_o) \right) dt \tag{8}$$

This paper also assumes that the green hotel supply chain members GSMHE and the OTA make decisions independently to maximize their own profit within an infinite time horizon. In summary, the net discounted profit function for the GSMHE and OTA can be obtained as:

$$\begin{cases} \max\limits_{S_h, E_h} \int_0^\infty e^{-\rho t} \left( \pi_h \cdot (\alpha G(t) + \beta W(t)) - \frac{\mu_1}{2} S_h^2 - \frac{\mu_2}{2} E_h^2 \right) dt \\ \max\limits_{E_o} \int_0^\infty e^{-\rho t} \left( \pi_o \cdot (\alpha G(t) + \beta W(t)) - \frac{\mu_3}{2} E_o^2 \right) dt \\ \text{s.t.} \dot{G}(t) = \tau_1 S_h(t) - \gamma G(t), G(0) = G_0 \\ \qquad \dot{W}(t) = \eta G(t) + \tau_2 E_h(t - d_h) + \tau_3 E_o(t - d_o) - \delta W(t), W(0) = W_0 \end{cases} \tag{9}$$

### 3.1 Online e-WOM promotion scenario

This scenario is hereafter denoted by O (online e-WOM). In this scenario, OTA makes efforts to promote e-WOM level online by its own. As the Stackelberg leader, OTA decides its online e-WOM effort $E_o^O$ first. Then the GSMHE chooses green service effort $S_h^O$. In this scenario $E_h^O = 0$, the decision of problems of the GSMHE and the OTA are respectively expressed as:

$$\max\limits_{S_h^O} \int_0^\infty e^{-\rho t} \left( \pi_h \cdot \left( \alpha G^O(t) + \beta W^O(t) \right) - \frac{\mu_1}{2} S_h^{O^2} \right) dt \tag{10}$$

$$\max_{E_o^O} \int_0^\infty e^{-\rho t} \left( \pi_o \cdot \left( \alpha G^O(t) + \beta W^O(t) \right) - \frac{\mu_3}{2} E_o^{O^2} \right) dt \tag{11}$$

For $G^O(t) > 0$ $W^O(t) > 0$, the maximal profits of the GSMHE and OTA at time t in the Hamilton-Jacobi-Bellman equation satisfy the following equations:

$$\rho V_h^O = \max_{S_h^O} \left\{ \begin{array}{l} \pi_h \left( \alpha G^O(t) + \beta W^O(t) \right) - \frac{\mu_1}{2} S_h^{O^2} + \frac{\partial V_h^O}{\partial G^O} \left( \tau_1 S_h^O(t) - \gamma G^O(t) \right) \\ + \frac{\partial V_h^O}{\partial W^O} \left( \eta G^O(t) + \tau_3 E_o^O(t - d_o) - \delta W^O(t) \right) \end{array} \right\} \tag{12}$$

$$\rho V_o^O = \max_{E_o^O} \left\{ \begin{array}{l} \pi_o \left( \alpha G^O(t) + \beta W^O(t) \right) - \frac{\mu_3}{2} E_o^{O^2} + \frac{\partial V_o^O}{\partial G^O} \left( \tau_1 S_h^O(t) - \gamma G^O(t) \right) \\ + \frac{\partial V_o^O}{\partial W^O} \left( \eta G^O(t) + \tau_3 E_o^O(t - d_o) - \delta W^O(t) \right) \end{array} \right\} \tag{13}$$

**Proposition 1** In the online e-WOM promotion scenario. The optimal green service $S_h^{O*}$ of the GSMHE, the optimal e-WOM effort $E_o^{O*}$ of the OTA are as follows:

$$\begin{cases} S_h^{O*} = \dfrac{\pi_h \left( \alpha \left( \rho + \delta \right) + \beta \eta \right)}{\left( \rho + \gamma \right) \left( \rho + \delta \right)} \cdot \dfrac{\tau_1}{\mu_1} \\ E_o^{O*} = \dfrac{\pi_o \beta}{\left( \rho + \delta \right)} \cdot \dfrac{\tau_3}{\mu_3} \cdot e^{\delta d_o} \end{cases} \tag{14}$$

The Proof is shown in the S1 File.

Substituting optimal solution Eq. (14) into Eq. (1) and (2) and solving for this first-order linear differential equation, yields the proposition 2.

**Proposition 2** In online e-WOM promotion scenario. The equilibrium green discourse level $G^{O*}$ and e-WOM level $W^{O*}$ is given by:

$$\begin{cases} G^{O*}(t) = G_\infty^O + \left( G_0 - G_\infty^O \right) e^{-\gamma t} \\ W^{O*}(t) = W_\infty^O + \frac{\eta}{\delta - \gamma} \left( G_0 - G_\infty^O \right) e^{-\gamma t} + \left( W_0 - W_\infty^O - \frac{\eta}{\delta - \gamma} \left( G_0 - G_\infty^O \right) \right) e^{-\delta t} \end{cases} \tag{15}$$

Where:

$$\begin{cases} G_\infty^O = \dfrac{\pi_h \left( \alpha \left( \rho + \delta \right) + \beta \eta \right)}{\left( \rho + \gamma \right) \left( \rho + \delta \right) \gamma} \cdot \dfrac{\tau_1^2}{\mu_1} \\ W_\infty^O = \dfrac{\pi_h \left( \alpha \left( \rho + \delta \right) + \beta \eta \right) \eta}{\left( \rho + \gamma \right) \left( \rho + \delta \right) \gamma \delta} \cdot \dfrac{\tau_1^2}{\mu_1} + \dfrac{\pi_o \beta}{\left( \rho + \delta \right) \delta} \cdot \dfrac{\tau_3^2}{\mu_3} \cdot e^{\delta d_o} \end{cases} \tag{16}$$

$G_\infty^O$, $W_\infty^O$ refer to the steady-state green discourse level and e-WOM level, respectively, when $t \to \infty$.

Substituting Eq. (14), (15) and (16) into Eq. (7) and (8), yields the optimal profit of GSMHE and OTA in online e-WOM promotion scenario:

$$\begin{aligned} J_h^{O*} = & \frac{\rho \pi_h \left( \alpha \gamma + \beta \eta \right) + \alpha \gamma \delta \pi_h}{\rho \left( \rho + \gamma \right) \left( \rho + \delta \right)} \left( \frac{\pi_h \left( \alpha \left( \rho + \delta \right) + \beta \eta \right)}{\left( \rho + \gamma \right) \left( \rho + \delta \right) \gamma} \cdot \frac{\tau_1^2}{\mu_1} \right) \\ & + \frac{\pi_h \beta}{\rho \left( \rho + \delta \right)} \left( \frac{\pi_h \left( \alpha \left( \rho + \delta \right) + \beta \eta \right) \eta}{\left( \rho + \gamma \right) \left( \rho + \delta \right) \gamma} \cdot \frac{\tau_1^2}{\mu_1} + \frac{\pi_o \beta}{\left( \rho + \delta \right)} \cdot \frac{\tau_3^2}{\mu_3} \cdot e^{\delta d_o} \right) \\ & + \frac{\pi_h \left( \alpha \left( \rho + \delta \right) + \beta \eta \right)}{\left( \rho + \gamma \right) \left( \rho + \delta \right)} G_0 + \frac{\pi_h \beta \alpha}{\rho + \delta} W_0 - \frac{\pi_h^2 \left( \alpha \left( \rho + \delta \right) + \beta \eta \right)^2}{2 \rho (\rho + \delta)^2 \left( \rho + \gamma \right)^2} \cdot \frac{\tau_1^2}{\mu_1} \end{aligned} \tag{17}$$

$$J_o^{O\star} = \frac{\rho \pi_o (\alpha \gamma + \beta \eta) + \alpha \gamma \delta \pi_o}{\rho (\rho + \gamma)(\rho + \delta)} \left( \frac{\pi_h (\alpha (\rho + \delta) + \beta \eta)}{(\rho + \gamma)(\rho + \delta) \gamma} \cdot \frac{\tau_1{}^2}{\mu_1} \right)$$
$$+ \frac{\pi_o \beta}{\rho (\rho + \delta)} \left( \frac{\pi_h (\alpha (\rho + \delta) + \beta \eta) \eta}{(\rho + \gamma)(\rho + \delta) \gamma} \cdot \frac{\tau_1{}^2}{\mu_1} + \frac{\pi_o \beta}{(\rho + \delta)} \cdot \frac{\tau_3{}^2}{\mu_3} \cdot e^{\delta d_o} \right)$$
$$+ \frac{\pi_o (\alpha (\rho + \delta) + \beta \eta)}{(\rho + \gamma)(\rho + \delta)} G_0 + \frac{\pi_o \beta \alpha}{\rho + \delta} W_0 - \frac{\pi_o{}^2 \beta^2}{2\rho(\rho + \delta)^2} \cdot \frac{\tau_3{}^2}{\mu_3} \cdot e^{2\delta d_o} \tag{18}$$

### 3.2 Online and offline e-WOM promotion cooperation scenario

This scenario is hereafter denoted by C (online and offline cooperation). In this scenario, GSMHE and OTA make efforts to promote e-WOM level respectively. As the Stackelberg leader, the OTA decides its online e-WOM effort $E_o^C$ first. Then the GSMHE chooses green service effort $S_h^C$ and its offline e-WOM effort $E_h^C$. the decision of problems of the GSMHE and the OTA are respectively expressed as

$$\max_{S_h^C, E_h^C} \int_0^\infty e^{-\rho t} \left( \pi_h \cdot \left( \alpha G^C(t) + \beta W^C(t) \right) - \frac{\mu_1}{2} S_h^{C2} - \frac{\mu_2}{2} E_h^{C2} \right) d t \tag{19}$$

$$\max_{E_o^C} \int_0^\infty e^{-\rho t} \left( \pi_o \cdot \left( \alpha G^C(t) + \beta W^C(t) \right) - \frac{\mu_3}{2} E_o^{C2} \right) d t \tag{20}$$

For $G^C(t) > 0$ $W^C(t) > 0$, the maximal profits of the GSMHE and OTA at time t in the Hamilton-Jacobi-Bellman equation satisfy the following equations:

$$\rho V_h^C = \max_{S_h^C, E_h^C} \left\{ \begin{array}{c} \pi_h \left( \alpha G^C(t) + \beta W^C(t) \right) - \frac{\mu_1}{2} S_h^{C2} - \frac{\mu_2}{2} E_h^{C2} \\ + \frac{\partial V_h^C}{\partial G^C} \left( \tau_1 S_h^C(t) - \gamma G^C(t) \right) + \frac{\partial V_h^C}{\partial W^C} \left( \begin{array}{c} \eta G^C(t) + \tau_2 E_o^C(t - d_h) \\ + \tau_3 E_o^C(t - d_o) - \delta W^C(t) \end{array} \right) \end{array} \right\} \tag{21}$$

$$\rho V_o^O = \max_{E_o^C} \left\{ \begin{array}{c} \pi_o \left( \alpha G^C(t) + \beta W^C(t) \right) - \frac{\mu_3}{2} E_o^{C2} + \frac{\partial V_o^C}{\partial G^C} \left( \tau_1 S_h^C(t) - \gamma G^C(t) \right) \\ + \frac{\partial V_o^C}{\partial W^C} \left( \eta G^C(t) \tau_2 E_o^C(t - d_h) + \tau_3 E_o^C(t - d_o) - \delta W^C(t) \right) \end{array} \right\} \tag{22}$$

**Proposition 3** In online offline e-WOM promotion cooperation scenario. The optimal green service $S_h^{C*}$ and offline e-WOM effort $E_h^{C*}$ of the GSMHE, the optimal online e-WOM effort $E_o^{C*}$ of the OTA are as follows:

$$\begin{cases} S_h^{C*} = \dfrac{\pi_h (\alpha (\rho + \delta) + \beta \eta)}{(\rho + \gamma)(\rho + \delta)} \cdot \dfrac{\tau_1}{\mu_1} \\ E_h^{C*} = \dfrac{\pi_h \beta}{(\rho + \delta)} \dfrac{\tau_2}{\mu_2} e^{\delta d_h} \\ E_o^{C*} = \dfrac{\pi_o \beta}{(\rho + \delta)} \dfrac{\tau_3}{\mu_3} e^{\delta d_o} \end{cases} \tag{23}$$

The Proof is shown in the S1 File.

Substituting optimal solution Eq. (23) into Eq. (1) and (2), solving for this first-order linear differential equation, yields the proposition 4.

**Proposition 4** In online offline e-WOM promotion cooperation scenario. The equilibrium green discourse level $G^{C*}$ and e-WOM level $W^{C*}$ are given by:

$$\begin{cases} G^{C^*}(t) = G_\infty^C + \left(G_0 - G_\infty^C\right) e^{-\gamma t} \\ W^{C^*}(t) = W_\infty^C + \dfrac{\eta}{\delta - \gamma}\left(G_0 - G_\infty^C\right) e^{-\gamma t} + \left(W_0 - W_\infty^C - \dfrac{\eta}{\delta - \gamma}\left(G_0 - G_\infty^C\right)\right) e^{-\delta t} \end{cases}$$

(24)

Where:

$$\begin{cases} G_\infty^C = \dfrac{\pi_h\left(\alpha\left(\rho + \delta\right) + \beta\eta\right)}{\left(\rho + \gamma\right)\left(\rho + \delta\right)\gamma} \cdot \dfrac{\tau_1{}^2}{\mu_1} \\ W_\infty^C = \dfrac{\pi_h\left(\alpha\left(\rho + \delta\right) + \beta\eta\right)\eta}{\left(\rho + \gamma\right)\left(\rho + \delta\right)\gamma\delta} \cdot \dfrac{\tau_1{}^2}{\mu_1} \\ \quad + \dfrac{\pi_h\beta}{\left(\rho + \delta\right)\delta} \cdot \dfrac{\tau_2{}^2}{\mu_2} \cdot e^{\delta d_h} + \dfrac{\pi_o\beta}{\left(\rho + \delta\right)\delta} \cdot \dfrac{\tau_3{}^2}{\mu_3} \cdot e^{\delta d_o} \end{cases}$$

(25)

$G_\infty^C$, $W_\infty^C$ refer to the steady-state green discourse level and e-WOM level, respectively, when $t \to \infty$.

Substituting Eq. (23), (24) and (25) into Eq. (19) and (20), yields the optimal profit of GSMHE and OTA in online offline e-WOM promotion cooperation scenario are:

$$\begin{aligned} J_h^{C^*} ={}& \frac{\rho\pi_h\left(\alpha\gamma + \beta\eta\right) + \alpha\gamma\delta\pi_h}{\rho\left(\rho + \gamma\right)\left(\rho + \delta\right)} \left(\frac{\pi_h\left(\alpha\left(\rho + \delta\right) + \beta\eta\right)}{\left(\rho + \gamma\right)\left(\rho + \delta\right)\gamma} \cdot \frac{\tau_1{}^2}{\mu_1}\right) \\ &+ \frac{\pi_h\beta}{\rho\left(\rho + \delta\right)} \left(\begin{array}{c}\frac{\pi_h\left(\alpha\left(\rho + \delta\right) + \beta\eta\right)\eta}{\left(\rho + \gamma\right)\left(\rho + \delta\right)\gamma} \cdot \frac{\tau_1{}^2}{\mu_1} + \frac{\pi_h\beta}{\left(\rho + \delta\right)} \frac{\tau_2{}^2}{\mu_2} e^{\delta d_h} \\ + \frac{\pi_o\beta}{\left(\rho + \delta\right)} \cdot \frac{\tau_3{}^2}{\mu_3} \cdot e^{\delta d_o}\end{array}\right) \\ &+ \frac{\pi_h\left(\alpha\left(\rho + \delta\right) + \beta\eta\right)}{\left(\rho + \gamma\right)\left(\rho + \delta\right)} G_0 + \frac{\pi_h\beta\alpha}{\rho + \delta} W_0 \\ &- \frac{\pi_h{}^2\left(\alpha\left(\rho + \delta\right) + \beta\eta\right)^2}{2\rho\left(\rho + \delta\right)^2\left(\rho + \gamma\right)^2} \frac{\tau_1{}^2}{\mu_1} - \frac{\pi_h{}^2\beta^2}{2\rho\left(\rho + \delta\right)^2} \frac{\tau_2{}^2}{\mu_2} e^{2\delta d_h} \end{aligned}$$

(26)

$$\begin{aligned} J_o^{C^*} ={}& \frac{\rho\pi_o\left(\alpha\gamma + \beta\eta\right) + \alpha\gamma\delta\pi_o}{\rho\left(\rho + \gamma\right)\left(\rho + \delta\right)} \left(\frac{\pi_h\left(\alpha\left(\rho + \delta\right) + \beta\eta\right)}{\left(\rho + \gamma\right)\left(\rho + \delta\right)\gamma} \cdot \frac{\tau_1{}^2}{\mu_1}\right) + \\ & \frac{\pi_o\beta}{\rho\left(\rho + \delta\right)} \left(\begin{array}{c}\frac{\pi_h\left(\alpha\left(\rho + \delta\right) + \beta\eta\right)\eta}{\left(\rho + \gamma\right)\left(\rho + \delta\right)\gamma} \cdot \frac{\tau_1{}^2}{\mu_1} + \frac{\pi_h\beta}{\left(\rho + \delta\right)} \frac{\tau_2{}^2}{\mu_2} e^{\delta d_h} \\ + \frac{\pi_o\beta}{\left(\rho + \delta\right)} \cdot \frac{\tau_3{}^2}{\mu_3} \cdot e^{\delta d_o}\end{array}\right) \\ &+ \frac{\pi_o\left(\alpha\left(\rho + \delta\right) + \beta\eta\right)}{\left(\rho + \gamma\right)\left(\rho + \delta\right)} G_0 + \frac{\pi_o\beta\alpha}{\rho + \delta} W_0 - \frac{\pi_o{}^2\beta^2}{2\rho\left(\rho + \delta\right)^2} \cdot \frac{\tau_3{}^2}{\mu_3} \cdot e^{2\delta d_o} \end{aligned}$$

(27)

### 3.3 Online and offline cost-sharing scenario

This scenario is hereafter denoted by S (cost sharing). Different with the cooperation scenario, the OTA decides its online e-WOM effort $E_o^S$ and cost sharing rate $\Phi$ first. Then the GSMHE chooses green service effort $S_h^S$ and its offline e-WOM effort $E_h^S$. The decision problems of the GSMHE and the OTA can be written as

$$\max_{S_h^S, E_h^S} \int_0^\infty e^{-\rho t} \left(\pi_h\left(\alpha G^S(t) + \beta W^S(t)\right) - \frac{\mu_1}{2} S_h^{S2} - \frac{\mu_2}{2}\left(1 - \Phi\right) E_h^{S2}\right) dt$$

(28)

$$\max_{E_o^S, \Phi, \ 0} \int_0^\infty e^{-\rho t} \left( \pi_0 \left( \alpha G^S(t) + \beta W^S(t) \right) - \frac{\mu_3}{2} E_o^{S2} - \frac{\mu_2}{2} \Phi E_h^{S2} \right) dt \tag{29}$$

For any W(t) > 0, the maximal profits of the GSMHE and the OTA at time t in the Hamilton-Jacobi-Bellman equation satisfy the following equations:

$$\rho V_h^S = \max_{S_h^S, E_h^S} \left\{ \begin{array}{c} \pi_h \left( \alpha G^S(t) + \beta W^S(t) \right) - \frac{\mu_1}{2} S_h^{S2} - \frac{\mu_2}{2} (1-\Phi) E_h^{S2} \\ + \frac{\partial V_h^S}{\partial G^S} \left( \tau_1 S_h^S(t) - \gamma G^S(t) \right) \\ + \frac{\partial V_h^S}{\partial W^S} \left( \eta G^S(t) + \tau_2 E_o^S(t-d_h) + \tau_3 E_o^S(t-d_o) - \delta W^S(t) \right) \end{array} \right\} \tag{30}$$

$$\rho V_o^S = \max_{E_o^S, \Phi,} \left\{ \begin{array}{c} \pi_0 \left( \alpha G^S(t) + \beta W^S(t) \right) - \frac{\mu_3}{2} E_o^{S2} - \frac{\mu_2}{2} \Phi E_h^{S2} \\ + \frac{\partial V_o^S}{\partial G^S} \left( \tau_1 S_h^S(t) - \gamma G^S(t) \right) \\ + \frac{\partial V_o^S}{\partial W^S} \left( \eta G^S(t) + \tau_2 E_o^S(t-d_h) + \tau_3 E_o^S(t-d_o) - \delta W^S(t) \right) \end{array} \right\} \tag{31}$$

**Proposition 5** In cost sharing scenario, the optimal green service $S_h^{S*}$ and offline e-WOM effort $E_h^{S*}$ of the GSMHE, the optimal online e-WOM effort $E_o^{S*}$ and cost sharing rate $\Phi^*$ of the OTA are as follows:

$$\begin{cases} S_h^{S*} = \frac{\pi_h \left( \alpha \left( \rho + \delta \right) + \beta \eta \right)}{\left( \rho + \gamma \right) \left( \rho + \delta \right)} \cdot \frac{\tau_1}{\mu_1} \\ E_h^{S*} = \frac{\pi_h \beta}{\left( \rho + \delta \right) \left( 1 - \Phi \right)} \frac{\tau_2}{\mu_2} e^{\delta d_h} \\ E_o^{S*} = \frac{\pi_0 \beta}{\left( \rho + \delta \right)} \frac{\tau_3}{\mu_3} e^{\delta d_o} \\ \Phi^* = \begin{cases} \frac{2\pi_0 - \pi_h}{2\pi_0 + \pi_h}, & \pi_0 \geq \frac{\pi_h}{2} \\ 0, & \pi_0 < \frac{\pi_h}{2} \end{cases} \end{cases} \tag{32}$$

The Proof is shown in the S1 File.

Substituting optimal solution Eq. (32) into Eq. (1) and (2), solving for this first-order linear differential equation, yields the proposition 6.

**Proposition 6** In cost sharing scenario. The equilibrium green discourse level $G^{S*}$ and e-WOM level $W^{S*}$ are given by:

$$\begin{cases} G^{S*}(t) = G_\infty^S + \left( G_0 - G_\infty^S \right) e^{-\gamma t} \\ W^{CS*}(t) = W_\infty^S + \frac{\eta}{\delta - \gamma} \left( G_0 - G_\infty^S \right) e^{-\gamma t} + \left( W_0 - W_\infty^S - \frac{\eta}{\delta - \gamma} \left( G_0 - G_\infty^S \right) \right) e^{-\delta t} \end{cases} \tag{33}$$

Where:

$$\begin{cases} G_\infty^S = \frac{\pi_h \left( \alpha \left( \rho + \delta \right) + \beta \eta \right)}{\left( \rho + \gamma \right) \left( \rho + \delta \right) \gamma} \cdot \frac{\tau_1^2}{\mu_1} \\ W_\infty^S = \frac{\pi_h \left( \alpha \left( \rho + \delta \right) + \beta \eta \right) \eta}{\left( \rho + \gamma \right) \left( \rho + \delta \right) \gamma \delta} \cdot \frac{\tau_1^2}{\mu_1} \\ + \frac{\pi_h \beta}{\left( 1 - \Phi \right) \left( \rho + \delta \right) \delta} \cdot \frac{\tau_2^2}{\mu_2} \cdot e^{\delta d_h} + \frac{\pi_0 \beta}{\left( \rho + \delta \right) \delta} \cdot \frac{\tau_3^2}{\mu_3} \cdot e^{\delta d_o} \end{cases} \tag{34}$$

$G^S_\infty$, $W^S_\infty$ refer to the steady-state green discourse level and e-WOM level, respectively, when $t \to \infty$.

Substituting Eq. (32), (33) and (34) into Eq. (28) and (29), yields the optimal profit of GSMHE and OTA in online offline e-WOM promotion cooperation scenario are:

$$
\begin{aligned}
J_h^{S\star} =\ & \frac{\rho\pi_h(\alpha\gamma + \beta\eta) + \alpha\gamma\delta\pi_h}{\rho(\rho+\gamma)(\rho+\delta)}\left(\frac{\pi_h(\alpha(\rho+\delta)+\beta\eta)}{(\rho+\gamma)(\rho+\delta)\gamma}\cdot\frac{\tau_1{}^2}{\mu_1}\right) \\
& + \frac{\pi_h\beta}{\rho(\rho+\delta)}\left(\begin{array}{c}\dfrac{\pi_h(\alpha(\rho+\delta)+\beta\eta)\eta}{(\rho+\gamma)(\rho+\delta)\gamma}\dfrac{\tau_1{}^2}{\mu_1}+\dfrac{\pi_h\beta}{(\rho+\delta)(1-\Phi)}\dfrac{\tau_2{}^2}{\mu_2}e^{\delta d_h}\\[2mm]+\dfrac{\pi_o\beta}{(\rho+\delta)}\dfrac{\tau_3{}^2}{\mu_3}e^{\delta d_o}\end{array}\right) \\
& + \frac{\pi_h(\alpha(\rho+\delta)+\beta\eta)}{(\rho+\gamma)(\rho+\delta)}G_0 + \frac{\pi_h\beta\alpha}{\rho+\delta}W_0 - \frac{\pi_h{}^2(\alpha(\rho+\delta)+\beta\eta)^2}{2\rho(\rho+\delta)^2(\rho+\gamma)^2}\frac{\tau_1{}^2}{\mu_1} \\
& - \frac{\pi_h{}^2\beta^2}{2\rho(\rho+\delta)^2(1-\Phi)}\frac{\tau_2{}^2}{\mu_2}e^{2\delta d_h}
\end{aligned}
\tag{35}
$$

$$
\begin{aligned}
J_o^{S\star} =\ & \frac{\rho\pi_o(\alpha\gamma+\beta\eta)+\alpha\gamma\delta\pi_o}{\rho(\rho+\gamma)(\rho+\delta)}\left(\frac{\pi_h(\alpha(\rho+\delta)+\beta\eta)}{(\rho+\gamma)(\rho+\delta)\gamma}\cdot\frac{\tau_1{}^2}{\mu_1}\right) \\
& + \frac{\pi_o\beta}{\rho(\rho+\delta)}\left(\begin{array}{c}\dfrac{\pi_h(\alpha(\rho+\delta)+\beta\eta)\eta}{(\rho+\gamma)(\rho+\delta)\gamma}\dfrac{\tau_1{}^2}{\mu_1}+\dfrac{\pi_h\beta}{(\rho+\delta)(1-\Phi)}\dfrac{\tau_2{}^2}{\mu_2}e^{\delta d_h}\\[2mm]+\dfrac{\pi_o\beta}{(\rho+\delta)}\dfrac{\tau_3{}^2}{\mu_3}e^{\delta d_o}\end{array}\right) \\
& + \frac{\pi_o(\alpha(\rho+\delta)+\beta\eta)}{(\rho+\gamma)(\rho+\delta)}G_0 + \frac{\pi_o\beta\alpha}{\rho+\delta}W_0 - \frac{\pi_o{}^2\beta^2}{2\rho(\rho+\delta)^2}\cdot\frac{\tau_3{}^2}{\mu_3}\cdot e^{2\delta d_o} \\
& - \frac{\pi_h{}^2\beta^2\Phi}{2\rho(\rho+\delta)^2(1-\Phi)^2}\frac{\tau_2{}^2}{\mu_2}e^{2\delta d_h}
\end{aligned}
\tag{36}
$$

## 4. Comparison of results and sensitivity analysis

### 4.1 Comparison of results

**Corollary 1** The relationship between the optimal green investment effort of GSMHE $S_h^*$, the optimal offline e-WOM efforts of GSMHE $E_h^*$ and the optimal e-WOM efforts of OTA $E_o^*$ under different scenarios are as follows:

$S_h^{S*} = S_h^{C*} = S_h^{O*}$.

$E_h^{S*} > E_h^{C*}$

$E_o^{S*} = E_o^{C*} = E_o^{O*}$

**Corollary 2** The ordinal relationship of the steady-state green discourse level $G_\infty^*$ and e-WOM level $W_\infty^*$ are as follows:

$G_\infty^{S*} = G_\infty^{C*} = G_\infty^{O*}$

$W_\infty^{S*} > W_\infty^{C*} > W_\infty^{O*}$

**Corollary 3** The relationship of optimal profit of GSMHE $J_h^*$ is shown as follows:

$J_h^{O*} < J_h^{C*} < J_h^{S*}$, when $d_h < \frac{\ln 2}{\delta}$

$J_h^{S*} < J_h^{C*} < J_h^{O*}$, when $d_h > \frac{\ln 2}{\delta}$

**Corollary 4** The relationship of optimal profit of OTA $J_o^*$ is shown as follows:

$J_o^{O*} < J_o^{C*} < J_o^{S*}$, when $d_h \leq \frac{\ln\frac{4\pi_o}{2\pi_o+\pi_h}}{\delta}$

$J_o^{O*} < J_o^{S*} < J_o^{C*}$, when $\frac{\ln\frac{4\pi_o}{2\pi_o+\pi_h}}{\delta} \leq d_h < \frac{\ln\frac{4\pi_o}{2\pi_o-\pi_h}}{\delta}$

$J_o^{S*} < J_o^{O*} < J_o^{C*}$, when $d_h > \frac{\ln\frac{4\pi_o}{2\pi_o-\pi_h}}{\delta}$

The Proofs of Corollary 1–4 are shown in the S1 File.

From corollaries 1, we observe that the green service efforts of GSMHEs and the e-WOM efforts of OTAs are consistent across the three scenarios. However, in the cost-sharing scenario, the e-WOM effort of GSMHEs exceeds that in the online and offline cooperation scenario. This suggests a stronger commitment to promoting offline word-of-mouth when OTA bear a portion of GSMHE's offline e-WOM promotion costs, highlighting the critical role of this cost-sharing arrangement in boosting GSMHEs' participation in e-WOM activities.

Corollary 2 indicates that, although the green discourse level remains constant across all scenarios, e-WOM levels exhibit a distinct pattern: they are highest in the cost-sharing scenario and lowest in the online e-WOM promotion scenario. This highlights the significance of coordinated strategies between OTAs and GSMHEs in maximizing e-WOM, particularly when the OTA shares part of the GSMHE's offline e-WOM promotion costs, which enables the achievement of the highest level of e-WOM.

Corollary 3 reveals that GSMHEs will engage in e-WOM promotion only when the time lag $d_h$ remains below a specific threshold $\frac{\ln 2}{\delta}$. In such cases, cost-sharing becomes the only optimal choice for maximizing profits once cooperation is initiated. This underscores the importance of timing in realizing returns from their offline e-WOM promotion efforts. As such, GSMHEs' participation in e-WOM promotion is highly dependent on the time delay remaining below this threshold. If the time lag $d_h$ exceeds the threshold, GSMHEs will likely opt out of cooperation and switch to the online e-WOM promotion mode, where the concept of $d_h$ no longer applies.

Corollary 4 shows that OTAs achieve their highest profits in the cost-sharing scenario when the time lag $d_h$ is below $\frac{\ln\frac{4\pi_o}{2\pi_o+\pi_h}}{\delta}$. This highlights that time lag $d_h$ significantly influences OTA profitability, with the benefits of cost-sharing diminishing as delays increase. However, when the time lag falls between $\frac{\ln\frac{4\pi_o}{2\pi_o+\pi_h}}{\delta}$ and $\frac{\ln\frac{4\pi_o}{2\pi_o-\pi_h}}{\delta}$ the cost-sharing scenario no longer provides the highest profits. Instead, the optimal profit for OTAs shifts to the online and offline cooperation scenario, with the online e-WOM promotion scenario yielding the lowest profits. Furthermore, when the time lag $d_h$ exceeds $\frac{\ln\frac{4\pi_o}{2\pi_o-\pi_h}}{\delta}$, the online and offline cooperation scenario provides the highest profit, followed by the independent e-WOM promotion scenario, with the cost-sharing scenario yielding the lowest profit.

This highlights the critical importance of delay timing for OTA profitability. While OTAs could potentially gain higher profits from the online and offline cooperation scenario when the time lag $d_h$ exceeds $\frac{\ln\frac{4\pi_o}{2\pi_o+\pi_h}}{\delta}$, GSMHEs will prioritize cost-sharing as their only optimal strategy for cooperation. As a result, the OTA is compelled to adopt the cost-sharing scenario. Moreover, since $\frac{\ln\frac{4\pi_o}{2\pi_o+\pi_h}}{\delta}$ is smaller than $\frac{\ln 2}{\delta}$, effective collaboration between OTAs and GSMHEs requires keeping time lag $d_h$ below $\frac{\ln\frac{4\pi_o}{2\pi_o+\pi_h}}{\delta}$. Adhering to this threshold is essential to ensure mutual benefits and sustain long-term cooperation.

### 4.2 Sensitivity analysis

**Corollary 5** For the optimal solutions of e-WOM promotion efforts of GSMHE and OTA, the green discourse and e-WOM trajectory with respect to key parameters:

(i)   $\frac{\partial S_h^\star}{\partial \tau_1} > 0, \frac{\partial S_h^\star}{\partial \alpha} > 0,\ \frac{\partial S_h^\star}{\partial \eta} > 0,\ \frac{\partial S_h^\star}{\partial \beta} > 0, \frac{\partial S_h^\star}{\partial \gamma} < 0, \frac{\partial S_h^\star}{\partial \delta} < 0$

(ii)  $\frac{\partial E_h^{C\star}}{\partial \tau_2} > 0, \frac{\partial E_h^{S\star}}{\partial \tau_2} > 0,\ \frac{\partial E_h^{C\star}}{\partial \beta} > 0,\ \frac{\partial E_h^{C\star}}{\partial \beta} > 0,\ \frac{\partial E_o^\star}{\partial \tau_3} > 0, \frac{\partial E_o^\star}{\partial \beta} > 0$

(iii) $\frac{\partial G_\infty^*}{\partial \tau_1} > 0, \frac{\partial G_\infty^*}{\partial \alpha} > 0,\ \frac{\partial G_\infty^*}{\partial \eta} > 0, \frac{\partial G_\infty^*}{\partial \beta} > 0,\ \frac{\partial G_\infty^*}{\partial \gamma} < 0, \frac{\partial G_\infty^*}{\partial \delta} < 0$

(iv)  $\frac{\partial W_\infty^*}{\partial \tau_1} > 0, \frac{\partial W_\infty^*}{\partial \tau_2} > 0, \frac{\partial W_\infty^*}{\partial \tau_3} > 0, \frac{\partial W_\infty^*}{\partial \alpha} > 0,\ \frac{\partial W_\infty^*}{\partial \eta} > 0, \frac{\partial W_\infty^*}{\partial \beta} > 0$

The Proofs of Corollary 5 are shown in the S1 File.

Corollary 5 suggests that as the contribution of green discourse to the e-WOM level increases, potential consumers exhibit a higher level of attention towards both the green discourse and e-WOM content embedded in Online Reviews (ORs). While its impact coefficient $\tau_1$ increases, GSMHE strives to enhance their green service efforts to achieve a higher

level of green discourse and e-WOM. When the green discourse and e-WOM decay rate increases, GSMHEs reduce their green service efforts, leading to a lower level of green discourse and e-WOM. Moreover, when potential consumers place greater emphasis on e-WOM, along with an increase in the impact coefficient $\tau_2, \tau_3$ GSMHE and OTA respond by improving their e-WOM promotion efforts resulting in higher level of e-WOM.

**Corollary 6** For the optimal solutions of e-WOM promotion efforts of GSMHE and OTA, the e-WOM trajectory, the optimal profit of GSMHE and OTA with respect to time lags:

(i) $\quad \frac{\partial E_h^{C\star}}{\partial d_h} > 0, \frac{\partial E_h^{S\star}}{\partial d_h} > 0, \frac{\partial E_o^{O\star}}{\partial d_o} = \frac{\partial E_o^{C\star}}{\partial d_o} = \frac{\partial E_o^{S\star}}{\partial d_o} > 0$

(ii) $\quad \frac{\partial W_\infty^*}{\partial d_h} > 0, \frac{\partial W_\infty^*}{\partial d_o} > 0$

(iii) $\quad \frac{\partial J_h^{C\star}}{\partial d_h} < 0, \frac{\partial J_h^{S\star}}{\partial d_h} < 0, \frac{\partial J_o^{C\star}}{\partial d_h} > 0, \frac{\partial J_o^{S\star}}{\partial d_h} > 0$

and when $d_h > \frac{\ln \frac{2\pi_o}{2\pi_o - \pi_h}}{\delta}, \frac{\partial J_o^{S\star}}{\partial d_h} < 0$, when $d_h < \frac{\ln \frac{2\pi_o}{2\pi_o - \pi_h}}{\delta}, \frac{\partial J_o^{S\star}}{\partial d_h} > 0$

(iv) $\quad \frac{\partial J_h^{O\star}}{\partial d_o} > 0, \frac{\partial J_h^{C\star}}{\partial d_o} > 0, \frac{\partial J_h^{S\star}}{\partial d_o} > 0, \frac{\partial J_o^{O\star}}{\partial d_o} < 0, \frac{\partial J_o^{C\star}}{\partial d_o} < 0, \frac{\partial J_o^{S\star}}{\partial d_o} < 0$

The Proofs of Corollary 6 are shown in the S1 File.

Corollary 6 demonstrates that as the time lags $d_h$ and $d_o$ increase, both GSMHE and OTA allocate greater efforts to e-WOM promotion individually, consequently enhancing the e-WOM level. Given that GSMHE and OTA bear the cost of e-WOM promotion themselves, an increase in the time lag $d_o$ leads to an improvement in the optimal profit of GSMHE while resulting in a decline in the optimal profit of OTA. Notably, in online and offline e-WOM cooperation scenario, an extended time lag $d_h$ reduces the profit of GSMHE but amplifies the profit of OTA. In the cost-sharing scenario, when $d_h$ is smaller than $\frac{\ln \frac{2\pi_o}{2\pi_o - \pi_h}}{\delta}$, the optimal profit of OTA exhibits an increasing trend as $d_h$ extends, but experiences a decline when $d_h$ surpasses $\frac{\ln \frac{2\pi_o}{2\pi_o - \pi_h}}{\delta}$.

While both time lags $d_h$ and $d_o$ influence e-WOM levels, an unbounded increase in these delays would be unrealistic, as it would lead to infinitely growing e-WOM. Instead, specific thresholds must be observed to ensure mutual benefits and optimal cooperation between GSMHE and OTA. To sustain the cost-sharing scenario, it is essential that the time lag $d_h$ stays below $\frac{\ln \frac{4\pi_o}{2\pi_o + \pi_h}}{\delta}$. Moreover, the time lag $d_o$ is subject to two constraints. It must not exceed $d_h$, and minimizing $d_o$ is essential for maximizing the OTA's profits, as its optimal profit decreases with increasing $d_o$.

## 5. Numerical analysis

To substantiate the comparison results and offer a clear exposition of the optimal strategies adopted by GSMHEs and OTAs, along with the changes in e-WOM in the three scenarios, we conducted numerical analysis. This empirical investigation aimed to provide additional managerial insights pertaining to optimal decision-making processes. In this section, the results of the previous analysis (such as the impact of time lag on members' e-WOM efforts and e-WOM level trajectory and the optimal profit of GSMHE and OTA) are further validated. Hence, under the premise of $\pi_o \geq \frac{\pi_h}{2}$, the related values are as follows.

$\tau_1 = 4, \tau_2 = 3, \tau_3 = 4, \eta = 3, \mu_1 = 4, \mu_2 = 3, \mu_3 = 3, \square = 0.3, \square = 0.5, \pi_h = 4, \pi_o = 6, G_0 = 1, W_0 = 10, \gamma = 0.3, \delta = 0.4, \rho = 0.9, \Phi = 0.5$.

Fig 2 illustrates that the e-WOM effort of GSMHE is higher in the cost-sharing scenario compared to the cooperation scenario. Furthermore, an increase in the lag time ($d_h$) results in a corresponding increase in GSMHE's e-WOM effort. The result is in line with the conclusion in corollary 1 and corollary 6. Fig 3 shows that the OTA's e-WOM efforts in three scenarios are equivalent. However, there exists a positive correlation between OTA's e-WOM efforts and the lag time ($d_o$), as higher lag time leads to increased e-WOM efforts by the OTA. The result is consistent with corollary 1 and corollary 6.

By setting the time lags $d_h$=1, $d_o$=0.5, we have defined the operational time span of the green hotel supply chain as $t \in [0, 30]$. The time trajectories of green discourse and e-WOM for the three scenarios are depicted in Fig 4 and Fig 5. It

is evident that the time trajectories of green discourse remain equivalent across all three scenarios. However, the time trajectory of e-WOM exhibits distinct patterns among the decision modes, consistently following the order $W^O < W^C < W^S$ which aligns with corollary 2. Moreover, the trajectories of green discourse and e-WOM in the three scenarios exhibit a monotonically increasing trend over time t, eventually converging to a steady state as time t approaches infinity.

The impact of $d_h$ and $d_o$ on the profit of GSMHE is depicted in Fig 6, with the time lag set as $d_h \in [0,3]$, $d_o \in [0,3]$. The findings reveal that the profit of GSMHE increases as the time lag $d_o$ increases, while it decreases with an increase in $d_h$, these results are in line with corollary 6. Importantly, in the cost-sharing scenario, GSMHE can achieve higher profits compared to cooperation and online promotion scenarios when the time lag $d_h$ is below a certain threshold $\frac{\ln 2}{\delta}$. However, in scenarios where the time lag $d_h$ exceeds the threshold, the profit in the online promotion scenario surpasses that of the other two scenarios. Consequently, GSMHE is compelled to cease offline e-WOM promotion once the time lag $d_h$ exceeds the threshold, thereby choosing the cooperation scenario is infeasible for GSMHE under any circumstances. These results align with the conclusions drawn in corollary 3. This result can be explained by the fact that as $d_h$ increases, GSMHE tends to invest more in offline e-WOM efforts, leading to a decline in their overall profit despite OTA subsidies for the offline e-WOM promotion costs. The result is in accordance with corollary 3.

The impact of $d_h$ and $d_o$ on the profit of OTA is depicted in Fig 7, with the time lag set as $d_h \in [0,3]$, $d_o \in [0,3]$. Fig 7 reveals a negative relationship between the OTA's profit and the increase in the time lag $d_o$, which can be attributed to the OTA's increasing investment in online e-WOM promotion as the time lag increases, which aligns with corollary 6. Moreover, Fig 7 demonstrates that among the three scenarios, the OTA achieves the highest profit in the cost-sharing scenario until the time lag $d_h$ surpasses a certain threshold $\ln\frac{\frac{4\pi_o}{2\pi_o+\pi_h}}{\delta}$. Although the online and offline cooperation scenario may yield higher profits for the OTA when the time lag $d_h$ exceeds threshold $\ln\frac{\frac{4\pi_o}{2\pi_o+\pi_h}}{\delta}$, it is important to note that GSMHE never opts for this scenario. As a result, the OTA must adhere to the cost-sharing scenario to optimize its profit. Since $\ln\frac{\frac{4\pi_o}{2\pi_o+\pi_h}}{\delta}$ is

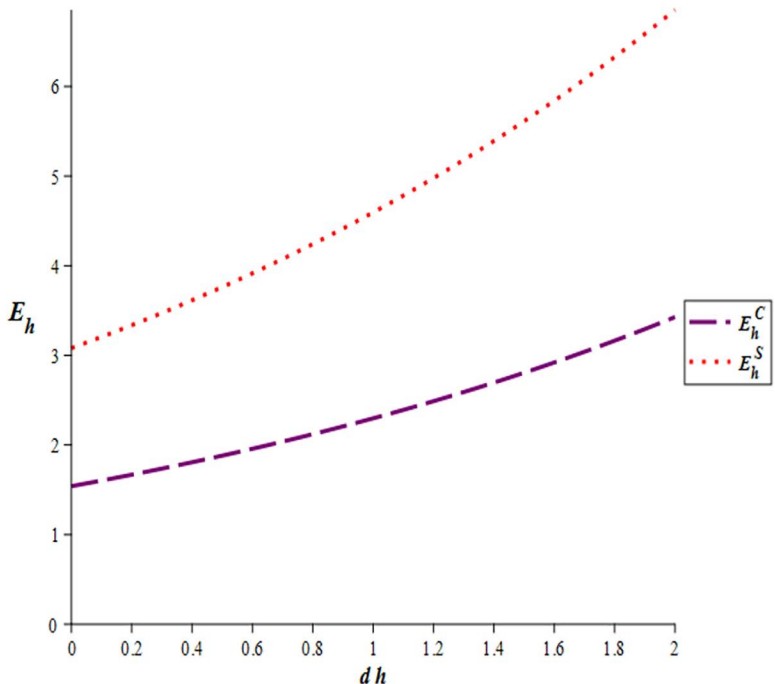

**Fig 2. Effects of time lags on GSMHE's e-WOM efforts.**

 

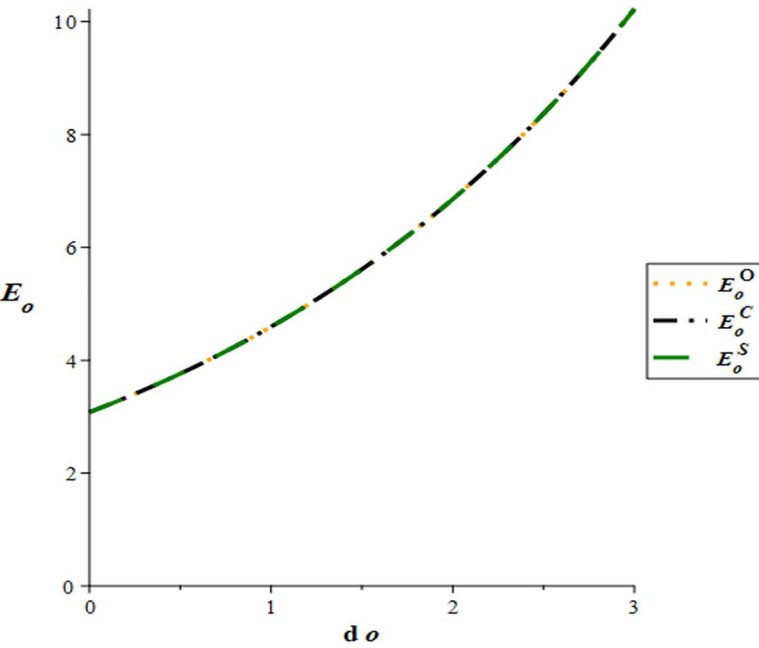

**Fig 3. Effects of time lags on OTA's e-WOM efforts.**

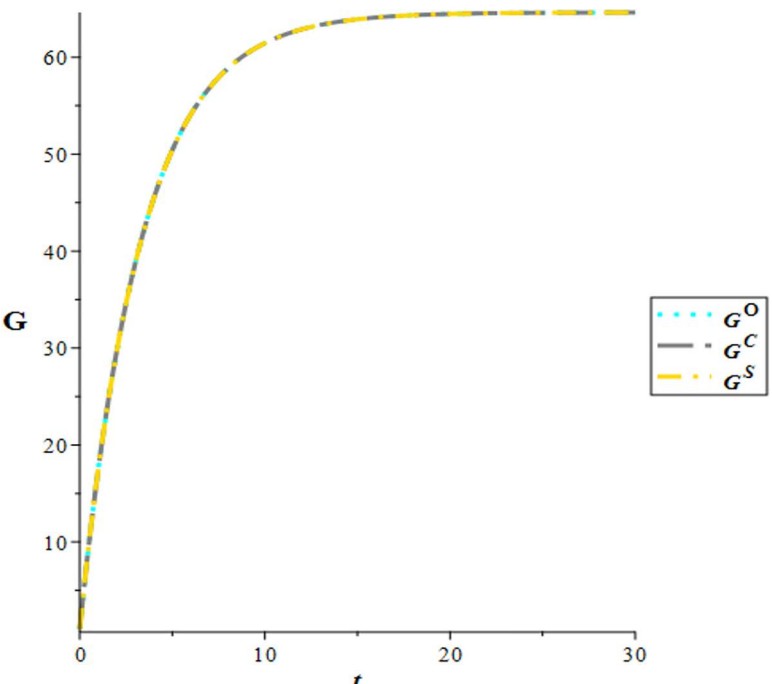

**Fig 4. Time trajectories of green discourse.**

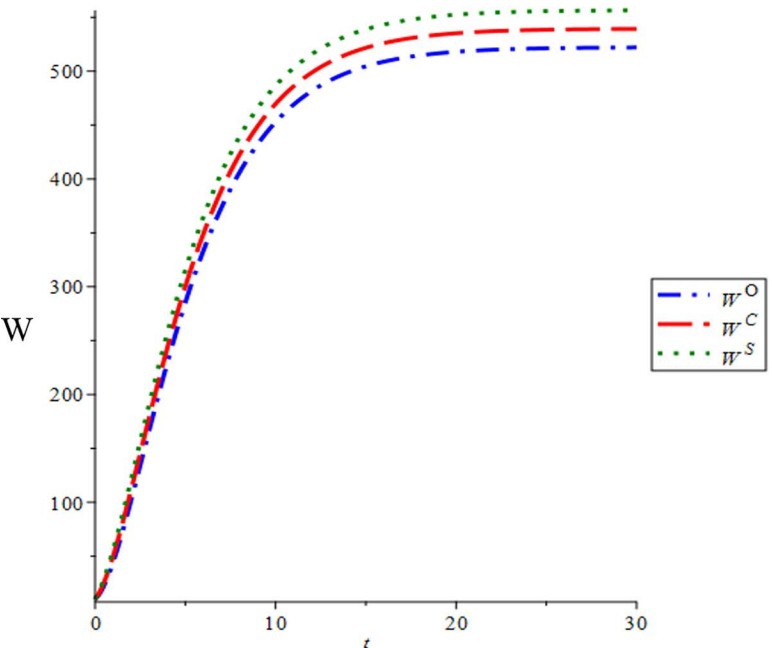

**Fig 5. Time trajectories of e-WOM.**

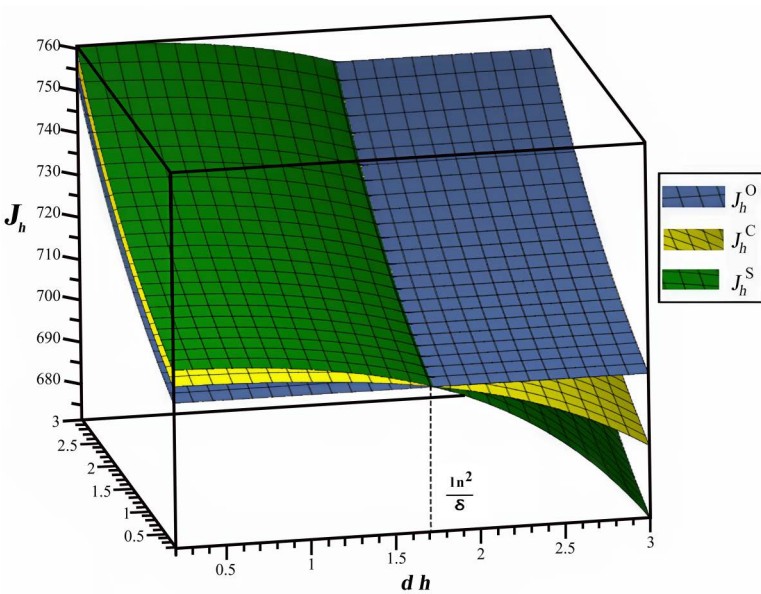

**Fig 6. Effects of time lags on GSMHE's performance.**

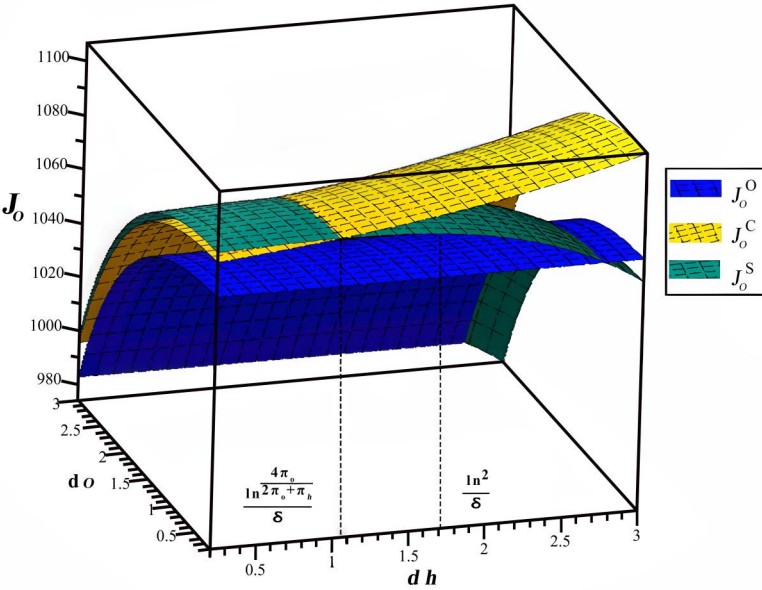

**Fig 7. Effects of time lags on OTA's performance.**

smaller than $\frac{\ln 2}{\delta}$, OTA and GSMHE should collaborate to keep $d_h$ remains within the threshold $\frac{\ln \frac{4\pi_o}{2\pi_o + \pi_h}}{\delta}$, which is consistent with corollary 4.

## 6 Conclusions and theoretical and managerial implications

### 6.1 Conclusion

Our study focuses on a single-channel hotel supply chain, comprising a GSMHE and an OTA. The GSMHE strategically allocates efforts towards green service, leading to a positive impact on green discourse. Both the GSMHE and OTA invest resources in e-WOM promotional activities, which contribute positively to the green hotel supply chain's e-WOM level. In addition, the effects of these investments are not immediate due to the inherent delay characteristics of online reviews posting. The market demand is influenced by both e-WOM and green discourse. Considering these assumptions, we explore and analyze three distinct scenarios: the online e-WOM promotion scenario, the online and offline e-WOM cooperation promotion scenario, and the cost-sharing scenario. Through rigorous analysis and comprehensive comparisons, our research yields some insightful findings:

1) The time lags of GSMHE's and OTA's efforts emerge as important factors in their decision-making processes. As $d_h$, $d_o$ increase, the GSMHE and OTA should strategically intensify their e-WOM efforts, respectively. However, these efforts are constrained by threshold limits on $d_h$, $d_o$. Moreover, the GSMHE's e-WOM efforts are found to be significantly higher in the cost-sharing scenario compared to the online and offline cooperation scenario, while the e-WOM efforts remain equal across all three scenarios.

2) The trajectories of both green discourse and e-WOM exhibit a consistent upward trend over time. Additionally, the trajectories of green discourse remain consistent across all three scenarios, while the e-WOM trajectory is observed to be the highest in the cost-sharing scenario.

3) Cost-sharing represents the optimal coordination model for both the GSMHE and OTA. The GSMHE does not select the cooperation scenario under any circumstances. While cost-sharing does not always yield the highest profit for the

GSMHE, the online promotion scenario becomes more profitable once the time lag $d_h$ exceeds certain threshold. For the OTA, its maximum profit is achieved through the cost-sharing scenario, but this is only the case until $d_h$ surpasses certain threshold. Although online and offline cooperation could potentially offer the OTA even higher profits beyond this point, the fact that the GSMHE never opts for this scenario means that both parties must focus on managing the time lag $d_h$ within this threshold. By simultaneously keeping the time lag $d_o$ as short as possible, the OTA and GSMHE can maximize their joint profits through the cost-sharing model.

## 6.2 Theoretical and managerial implications

In terms of theoretical contributions, our study further provides the evident supporting the impact of e-WOM and green discourse on the demand for the green hotel supply chain. Previous studies on the dynamic relationship of green tourism supply chains under a differential game approach have predominantly concentrated on factors such as price and advertising [89–91], neglecting the significant role of e-WOM at the form of ORs. However, in the era of e-commerce, the accessibility and convenience of obtaining ORs have increased, making them a trusted source of information for travelers when selecting hotel [92].

Moreover, the inclusion of green discourse within ORs has been proved to enhance the e-WOM of hotels. As consumers place greater importance on sustainability and environmental consciousness, the presence of positive green discourse in ORs positively impacts their perception of the hotel's green initiatives [69]. This, in turn, influences consumers' evaluation of e-WOM level, making it a crucial factor in shaping e-WOM and success of green hotel supply chains [62].

Drawing from our analyses and conclusions, several practical implications can be derived for green hotel supply chain management. Firstly, ORs serve as a form of e-WOM for green hotel supply chains, and the integration of green discourse within ORs plays an important role in enhancing the overall e-WOM level. Recognizing this, OTA should strategically invest in online e-WOM promotion efforts to directly improve the e-WOM level. Additionally, GSMHE should focus on investing in green service efforts to cultivate positive green discourse within ORs while investing offline e-WOM promotion efforts to improve e-WOM level.

Furthermore, GSMHE and OTA should coordinate their efforts to solicit reviews from guests at different stages of their interaction with the hotel. GSMHE can focus on gathering reviews during guests' stay by actively interacting with them and encouraging them to share their experiences through reviews. Meanwhile, OTA can follow up after guests' departure by conducting online e-WOM promotion, utilizing various online platforms to reach out to guests and prompt them to post reviews.

Secondly, the optimal decision-making method for a green hotel supply chain should consider the threshold of the GSMHE's time lag$d_h$. If the time lag falls within this threshold, the cost-sharing decision can be deemed suitable, ensuring that both GSMHEs and OTAs can maximize their profitability. Furthermore, as a leading role in the green hotel supply chain, OTAs should actively encourage GSMHEs to engage in offline e-WOM promotion by sharing the offline e-WOM promotion costs, fostering mutual benefit and driving overall profitability. In addition, it is crucial for GSMHE and OTA to monitor the duration of guest review postings and effectively manage and minimize these time lags for achieving a harmonious equilibrium between attaining a favorable e-WOM level and preserving optimal profitability.

While this study makes valuable contributions, it has some limitations. First, we focus on the impact of e-WOM and green discourse on demand but exclude price factors, which play a key role in consumer decision-making. Future research could explore how price interacts with e-WOM and green initiatives. Second, we assume uniform marginal revenues for GSMHE and OTA in the profit function, but these may vary depending on the platform, cooperation model, and other factors. Future studies should account for these differences to better understand profit dynamics in the green hotel supply chain.

## Supporting information

**S1 File. Proofs.**
(DOCX)

## Acknowledgments

We appreciate the editor and anonymous reviewers for their valuable contributions to improving the quality of the original version of this work.

## Author contributions

**Conceptualization:** Lujie Hao.

**Funding acquisition:** Lujie Hao.

**Methodology:** Bingkun Lin, Yaping Zhu.

**Supervision:** Bingkun Lin.

**Writing – original draft:** Lujie Hao, Yaping Zhu.

**Writing – review & editing:** Bingkun Lin.

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
