## [Decision Letter · Decision Letter 0]

Dear Dr. Bingkun Lin

Thank you for submitting your manuscript to PLOS ONE. After careful consideration, we feel that it has merit but does not fully meet PLOS ONE’s publication criteria as it currently stands. Therefore, we invite you to submit a revised version of the manuscript that addresses the points raised during the review process.

We look forward to receiving your revised manuscript.

Kind regards,

Yasuko Kawahata

Academic Editor

PLOS ONE

Journal Requirements:

“LH acknowledges the support from the Social Science Foundation of Fujian (grant number FJ2023B059) and the Fuzhou Key Research Institute of Social Science Foundation (grant number 2024FZB032).The funders had no role in study design, data collection and analysis, decision to publish, or preparation of the manuscript.”

Reviewers' comments:

Reviewer's Responses to Questions

**Comments to the Author**

1. Is the manuscript technically sound, and do the data support the conclusions?

Reviewer #1: Yes

Reviewer #2: Yes

2. Has the statistical analysis been performed appropriately and rigorously?

Reviewer #1: Yes

Reviewer #2: Yes

3. Have the authors made all data underlying the findings in their manuscript fully available?

Reviewer #1: Yes

Reviewer #2: Yes

4. Is the manuscript presented in an intelligible fashion and written in standard English?

Reviewer #1: Yes

Reviewer #2: Yes

Reviewer #1: This paper examines the dynamic impacts of E-WOM and green discourse on green hotel supply chain performance with time lag effect. Specifically, it employs differential game method to establish models under three scenarios, i.e., online e-WOM promotion scenario, online and offline e-WOM promotion cooperation scenario and online and offline cost-sharing scenario. Then the comparisons across the three scenarios are conducted and numerical examples are provided. From my perspective, the topic is meaningful, the models are well designed and some managerial insights are derived. It deserves publication after minor revisions. My concerns are as follows:

1. In Section 2, please supplement the literature related to OTA and E-WOM published from 2023 to mid-2024.

2. There is lack of sufficient analysis for Corollary 1 to Corollary 4. Thus, I suggest the authors strengthen this part.

3. There are some grammar and spelling mistakes in the paper. For example, in Line 201, “notation” should be revised as “notations”; in Line 272, “impact” should be revised as “impacts”; in Line 283, “cost” should be revised as “costs”; in Line 440, “green disclosure” should be revised as “green discourse”. Hence, I suggest the authors check and revise the text of the whole manuscript carefully.

Reviewer #2: Dynamic Impact of E-WOM and Green Discourse on Green Hotel Supply Chain Performance with Time Lag Effect (PONE-D-24-32543)

This paper explores an important topic in hotel supply chain management by analyzing an online green hotel supply chain consisting of a GSMHE and an OTA. However, there are some shortcomings in this paper. Specifically, I have the following concerns:

(1) The introduction section mentions that some hotels have gained a competitive advantage by actively adopting and implementing environmental protection measures, but it does not provide real-life cases corresponding to the discussion in this paper, resulting in a lack of sufficient empirical support for the research context.

(2) The introduction section lacks a listing of the innovations of the paper.

(3) Although the title of the two parts of the literature review indicates that the second part focuses more on “green” related content, the actual content of the first part also covers “green” and “sustainable” topics, resulting in some overlap between the two parts. However, in reality, the first part also covers topics such as “green” and “sustainable”, resulting in a lack of clarity in categorization between the two parts and some overlap in content. In addition, the literature review section is currently characterized by the phenomenon of stacking of literature and lack of in-depth analysis and review. It is recommended that critical thinking of the literature be strengthened, and the contributions and limitations of the studies and their relevance to the current research theme be explored in depth, so as to enhance the academic depth and theoretical value of the literature review.

(4) This paper focuses on the dynamic impacts under delayed effects; however, the literature review section does not sufficiently address the current state of research related to this topic. It is recommended to include a systematic review of existing studies on delayed effects and their dynamic impacts, in order to highlight the progress, limitations, and gaps in current research, as well as to clarify the contributions of this paper.

(5) In the Model formulation section, the authors only include non-price factors in the consideration of the demand function, however, in real situations, price is usually a key factor influencing consumer decisions. It is recommended that the authors add the specific reasons for not considering the price factor to enhance the rationality and explanatory power of the model construction.

(6) In the Model formulation section, the marginal revenues of GSMHE and OTA are set to a uniform value in the profit function. However, in reality, the marginal revenues of different platforms may differ significantly. It is recommended that the authors add the theoretical or practical rationale behind this setting.

(7) In section 4.1 Comparison of results, the current textual description is limited to a simple statement of the comparative results and lacks in-depth analysis, which is recommended to be expanded.

(8) The layout of the article's figures and tables is not very aesthetically pleasing, and changes are recommended.

(9) Corollary 6 argues that as the delay time increases, both GSMHE and OTA should each allocate a greater effort to promote e-WOM and, therefore, increase the e-WOM level. From this conclusion, if the delay time grows infinitely, the e-WOM level will keep increasing. Obviously, this conclusion does not correspond to reality and the authors do not develop a rational explanation for this conclusion.

**Do you want your identity to be public for this peer review?** For information about this choice, including consent withdrawal, please see our Privacy Policy

Reviewer #1: No

Reviewer #2: No

---

## [Author Response · Author response to Decision Letter 1]

31 Mar 2025

Response to Reviewers

Manuscript Title: Dynamic Impact of E-WOM and Green Discourse on Green Hotel Supply Chain Performance with Time Lag Effect

Manuscript ID: PONE-D-24-32543

Dear Editor and Reviewers,

We sincerely appreciate the time and effort that you and the reviewers have devoted to reviewing our manuscript. We are grateful for the valuable and constructive comments, which have helped us improve the quality and clarity of our work.

We have carefully considered all the comments and suggestions and made corresponding revisions in the manuscript. Below, we provide a point-by-point response to each comment. All changes in the revised manuscript are marked in yellow, and we have revised the manuscript accordingly.

Reviewer #1:

Comment 1:

In Section 2, please supplement the literature related to OTA and E-WOM published from 2023 to mid-2024.

Response:

Thank you very much for your valuable suggestion. In response, we have thoroughly reviewed recent literature and incorporated a series of relevant and up-to-date studies on OTA and E-WOM published between 2023 and -2025. Specifically, we have added references numbered 41, 47, 50, 51, 53, 55, 57, 63, 66, 67, 68, 69, 70, 71, 73, 77, 79, 81, and 82 to Section 2 of the revised manuscript. These additions enrich the theoretical foundation and ensure the literature review reflects the latest advancements in the field.

Comment 2:

There is lack of sufficient analysis for Corollary 1 to Corollary 4. Thus, I suggest the authors strengthen this part.

Response:

Thank you for your valuable suggestion. In response, we have added an in-depth analysis of the strategic differences across the three cooperation scenarios in Lines 413–452, Pages 28–30 of the revised manuscript. In the analysis of Corollaries 1 to 4, we not only compared the behavioral outcomes of green service efforts and e-WOM efforts under three different scenarios, but also explored the critical thresholds that determine whether cooperation between GSMHEs and OTAs is feasible and beneficial, particularly in Corollary 3 and Corollary 4. These thresholds provide important managerial insights by identifying the conditions under which cost-sharing becomes the dominant strategy, and by highlighting the critical threshold of the time lag\ d_h that both parties should strive to keep below to sustain effective cooperation under the cost-sharing scenario. We believe these additions strengthen the theoretical contribution and practical relevance of our study.

Comment 3:

There are some grammar and spelling mistakes in the paper. For example, in Line 201, “notation” should be revised as “notations”; in Line 272, “impact” should be revised as “impacts”; in Line 283, “cost” should be revised as “costs”; in Line 440, “green disclosure” should be revised as “green discourse”. Hence, I suggest the authors check and revise the text of the whole manuscript carefully.

Response:

Thank you for your careful reading and helpful comments. We have thoroughly revised the manuscript to correct the grammar, spelling, and word usage issues you pointed out. In addition, we conducted a comprehensive review of the entire manuscript to ensure clarity, readability, and consistency throughout.

Reviewer #2:

Comment 1:

The introduction section mentions that some hotels have gained a competitive advantage by actively adopting and implementing environmental protection measures, but it does not provide real-life cases corresponding to the discussion in this paper, resulting in a lack of sufficient empirical support for the research context.

Response:

Thank you for your insightful comment. In response, we have added a real-life case in Lines 29–30, Page 3 of the revised manuscript to enhance the empirical support for our research context. Specifically, we introduced The Grand Hotel in Birmingham, which has successfully implemented eco-friendly practices, thereby attracting more sustainability-conscious travelers. This example illustrates how green initiatives can contribute to a hotel's competitive advantage and supports the relevance of our study.

Comment 2:

The introduction section lacks a listing of the innovations of the paper.

Response:

Thank you for your valuable suggestion. In response, we have elaborated on the innovations of the paper in Lines 55–66, Page 5 of the revised manuscript. To highlight the novelty of our approach, we used phrases such as "introduces a novel approach" and "a factor that has not been fully addressed in previous studies" to emphasize the key contributions of our research. Due to the overall structure of the paper, we chose not to present the innovations in a listing format, but rather integrated them seamlessly within the narrative to maintain readability and flow.

Comment 3:

Although the title of the two parts of the literature review indicates that the second part focuses more on “green” related content, the actual content of the first part also covers “green” and “sustainable” topics, resulting in some overlap between the two parts. However, in reality, the first part also covers topics such as “green” and “sustainable”, resulting in a lack of clarity in categorization between the two parts and some overlap in content. In addition, the literature review section is currently characterized by the phenomenon of stacking of literature and lack of in-depth analysis and review. It is recommended that critical thinking of the literature be strengthened, and the contributions and limitations of the studies and their relevance to the current research theme be explored in depth, so as to enhance the academic depth and theoretical value of the literature review.

Response:

Thank you for your thoughtful comment. In response, we have restructured the Introduction and Literature Review sections to improve clarity and reduce redundancy. Specifically, we streamlined the Introduction to better highlight the research problem and moved some parts to the Literature Review section, thereby minimizing overlap between the two parts. Additionally, we revised the literature review to enhance coherence between its subsections and clarify their respective focuses. To strengthen academic depth, we incorporated more critical thinking into our discussion in Lines 158–170 and 179–184, Pages 10–12 of the revised manuscript, where we provide critical evaluations of existing literature, highlight their contributions and limitations, and explain how they relate to the present study.

Comment 4:

This paper focuses on the dynamic impacts under delayed effects; however, the literature review section does not sufficiently address the current state of research related to this topic. It is recommended to include a systematic review of existing studies on delayed effects and their dynamic impacts, in order to highlight the progress, limitations, and gaps in current research, as well as to clarify the contributions of this paper.

Response:

Thank you for your insightful comment. In response, we have expanded the discussion of delayed effects in the Literature Review section (Pages 11–12, Lines 171–184). Based on our explanation of the causes of e-WOM time lag, we conducted a focused review of relevant literature on time delays and their dynamic impacts in supply chain research. Specifically, we summarized prior studies addressing delay effects in areas such as Corporate Social Responsibility (CSR) investments, low-carbon investment, green technology spillovers, product and service quality, and advertising effects. We critically reviewed their findings to highlight their relevance and limitations in the context of our study. Due to space constraints and to maintain narrative coherence, we did not include a separate subsection title for this content, but we believe the integration enhances the depth and relevance of the literature review.

Comment 5:

In the Model formulation section, the authors only include non-price factors in the consideration of the demand function, however, in real situations, price is usually a key factor influencing consumer decisions. It is recommended that the authors add the specific reasons for not considering the price factor to enhance the rationality and explanatory power of the model construction.

Response:

Thank you for your thoughtful comment. We agree that price is an important factor influencing consumer decisions in real-world scenarios. To clarify our modeling choice, we have added an explanation in Page 17, Lines 262–266, stating: "To maintain a clear focus on the impact of these non-price factors, we exclude price considerations from the model, in accordance with the demand function formulation used in similar research [89]." We also cite Reference [89] as an example to support the rationale for our modeling approach. Furthermore, we have acknowledged this modeling simplification as one of the study’s limitations and provided suggestions for future research directions in Page 42, Lines 656–659. We hope these revisions help enhance the transparency and academic rigor of our model formulation.

Comment 6:

In the Model formulation section, the marginal revenues of GSMHE and OTA are set to a uniform value in the profit function. However, in reality, the marginal revenues of different platforms may differ significantly. It is recommended that the authors add the theoretical or practical rationale behind this setting.

Response:

Thank you for your insightful comment. We sincerely acknowledge that the assumption of uniform marginal revenue for GSMHEs and OTAs is a simplification that may not adequately capture the diversity of real-world platforms. In actual practice, marginal revenues are likely to vary due to differences in the platform, cooperation model, and other factors. This is indeed a valuable point that we had not fully taken into account in our initial model formulation.

While we were not able to incorporate a more differentiated treatment of marginal revenues in the current version due to the complexity it would introduce, we have added a discussion of this limitation in the concluding section (Page 42, Lines 659–663). We also highlight this as a meaningful direction for future research, which could further enhance the realism and applicability of the model. We truly appreciate this constructive suggestion, which helps us improve the rigor of our work.

Comment 7:

In section 4.1 Comparison of results, the current textual description is limited to a simple statement of the comparative results and lacks in-depth analysis, which is recommended to be expanded.

Response:

Thank you very much for your thoughtful suggestion. We sincerely appreciate your feedback on the need for a more in-depth analysis in Section 4.1. In response, we have significantly expanded this section in the revised manuscript (Pages 28–30, Lines 413–452). We have provided a more detailed discussion of the strategic differences across the three cooperation scenarios.

Specifically, in the analysis of Corollaries 1 to 4, we not only compared the behavioral outcomes of green service efforts and e-WOM efforts in different scenarios, but also examined the critical thresholds that determine whether cooperation between GSMHEs and OTAs is feasible and advantageous. In particular, Corollaries 3 and 4 highlight the importance of specific time lag thresholds d_h that both parties should aim to control in order to sustain effective cooperation under the cost-sharing scenario. These insights add substantial managerial value by identifying the conditions under which cost-sharing becomes the most effective strategy.

We hope these revisions address your concerns and further enhance the theoretical and practical relevance of our study. Thank you again for your valuable input.

Comment 8:

The layout of the article's figures and tables is not very aesthetically pleasing, and changes are recommended.

Response:

Thank you for your valuable feedback regarding the layout of the figures and tables. In response, we have made optimizations to the arrangement and positioning of the figures and tables to improve the overall visual presentation. The previous placement of some figures was not ideal, and we have adjusted them to enhance readability and flow. These layout adjustments aim to make the manuscript more aesthetically pleasing and easier to follow. We hope these changes meet your expectations and improve the overall presentation.

Comment 9:

Corollary 6 argues that as the delay time increases, both GSMHE and OTA should each allocate a greater effort to promote e-WOM and, therefore, increase the e-WOM level. From this conclusion, if the delay time grows infinitely, the e-WOM level will keep increasing. Obviously, this conclusion does not correspond to reality and the authors do not develop a rational explanation for this conclusion.

Response:

Thank you for your insightful comment. We acknowledge the concern regarding the unrealistic implication that e-WOM levels would increase indefinitely with the growing delay time. In response, we have provided a more realistic and rational explanation in the revised manuscript (Page 33, Lines 490–496). Specifically, we clarify that, in reality, the time lags d_h and d_o cannot increase indefinitely. d_h has an upper threshold beyond which it loses its relevance, and d_o must not exceed d_h, with smaller values of d_o being more optimal.

Once again, we truly appreciate your time and the reviewers' detailed and helpful feedback. We hope that the revisions meet your expectations and that the revised manuscript is now suitable for publication in your esteemed journal.

Sincerely,

---

## [Decision Letter · Decision Letter 1]

Dynamic Impact of E-WOM and Green Discourse on Green Hotel Supply Chain Performance with Time Lag Effect

PONE-D-24-32543R1

Dear Dr. Bingkun Lin,

We’re pleased to inform you that your manuscript has been judged scientifically suitable for publication and will be formally accepted for publication once it meets all outstanding technical requirements.

Kind regards,

Yasuko Kawahata

Academic Editor

PLOS ONE

Additional Editor Comments (optional):

Reviewers' comments:

Reviewer's Responses to Questions

**Comments to the Author**

Reviewer #1: All comments have been addressed

Reviewer #2: All comments have been addressed

2. Is the manuscript technically sound, and do the data support the conclusions?

Reviewer #1: Yes

Reviewer #2: Yes

3. Has the statistical analysis been performed appropriately and rigorously?

Reviewer #1: Yes

Reviewer #2: Yes

4. Have the authors made all data underlying the findings in their manuscript fully available?

Reviewer #1: Yes

Reviewer #2: Yes

5. Is the manuscript presented in an intelligible fashion and written in standard English?

Reviewer #1: Yes

Reviewer #2: Yes

Reviewer #1: By review the revised edition carefully�I find that all my concerns for the original edition have been well addressed, and the quality of the manuscript have also been greatly improved based on other reviewers' comments. Hence, I recommend it be accepted and published.

Reviewer #2: The author has carefully revised the manuscript according to the review comments, therefore, I suggest accepting the manuscript for publication.

**Do you want your identity to be public for this peer review?** For information about this choice, including consent withdrawal, please see our Privacy Policy

Reviewer #1: No

Reviewer #2: No

---

## [Editor Report · Acceptance letter]

PONE-D-24-32543R1

PLOS ONE

Dear Dr. Lin,

I'm pleased to inform you that your manuscript has been deemed suitable for publication in PLOS ONE. Congratulations! Your manuscript is now being handed over to our production team.

Kind regards,

on behalf of

Dr. Yasuko Kawahata

Academic Editor

PLOS ONE